# The role of livelihood diversification in agricultural modernization among tribal farmers in Bangladesh: A binary logistic model approach

**Anika Tasnim**[1], **Jasim Uddin Ahmed**[1], **Md. Jahangir Alam Khan**[2], **Md. Abdullah Al-Noman** [ID][3], **Arifa Jannat** [ID][4], **Md. Monirul Islam** [ID][3,5]*

**1** Department of Agricultural Economics and Policy, Sylhet Agricultural University, Sylhet, Bangladesh, **2** Dhaka School of Economics, University of Dhaka, Dhaka, Bangladesh, **3** Department of Agricultural Economics, Bangladesh Agricultural University, Mymensingh, Bangladesh, **4** Institute of Agribusiness and Development Studies, Bangladesh Agricultural University, Mymensingh, Bangladesh, **5** Commonwealth Scientific and Industrial Research Organisation—CSIRO, Waite Campus, Adelaide, South Australia, Australia

* monir96@bau.edu.bd

## Abstract

This study aimed to evaluate the influence of livelihood diversification on tribal farmers' decision to adopt modern agricultural practices, while also examining the profitability of various agricultural practices and identifying key challenges and policy recommendations for enhancing the sustainability of tribal farming livelihoods in Bangladesh. Data were collected through a structured questionnaire during the month of February 2021 from 115 tribal farmers, followed by a simple random sampling technique. In the study area, only 9.57% of farmers were involved in crop cultivation. The study calculated that human labor cost was the highest in rice production (about 37%). The benefit-cost ratio (BCR) of rice production, livestock rearing, and homestead gardening was estimated at 1.46, 1.41, and 1.09, respectively. Among all agricultural components practiced by the tribal farmers, rice production was more efficient than others. The binary logistic regression findings showed that livelihood diversification, age of the farmers, access to information, house-hold income, and farming experience had a significant influence on the adoption decision of modern agricultural practices. Therefore, promoting livelihood diversification among tribal farmers is crucial for enhancing their resilience and income sustainability. This could include providing training, access to markets, and financial support for activities such as poultry and livestock rearing and non-farm enterprises like homestead gardening.

## Introduction

Indigenous people are uniquely identified because of their distinct culture, rites and rituals, dialect, and dresses from the larger community with whom they coexist. According to the United Nations International Children's Emergency Fund [1] 1.75 million indigenous people are living in Bangladesh whereas Chattogram has the highest concentration of indigenous

**Data availability statement:** All relevant data are within the manuscript and its Supporting Information files.

**Funding:** The author(s) received no specific funding for this work.

**Competing interests:** The authors have declared that no competing interests exist.

people. The hilly areas of Chattogram are called Chattogram Hill Tracts (CHT), where education, communication, and medical facilities are rare, as well as poverty and illiteracy rates are high [2]. Indigenous people are mainly involved in agriculture, daily wage labor, and petty trade. Rozario et al. [3] mentioned that these indigenous peoples in southeastern Bangladesh are highly vulnerable to natural calamities because of their poor livelihood standards. They have limited access to electricity, pure drinking water, and sanitary facilities [4]. Being one of the fastest-growing economies, Bangladesh has decreased its extreme poverty level from 82% in 1972 to 12.9% in 2016 [5]. However, this incredible achievement rarely touched this group of people due to a lack of accessibility and fewer opportunities to engage in alternative income-generating activities (IGAs) [6].

One of the most effective ways to fight poverty is to diversify one's sources of income, which involves moving from farm to non-farm activities as the rural non-farm economy grows [7]. Rural households undertake this process, termed rural livelihood diversification, to bolster their resilience and quality of life [7,8]. It is worth noting that Indigenous peoples in the CHT are predominantly shifting cultivators [9], and almost 70% rely on agriculture as a major or secondary source of income [10], but their livelihood practices have varied slightly over time. There are two major types of cropland in CHT, those are plough and *jhum (Jhum is a method of cultivation of different crops with rice on a hill slope. It is also known as the slash-and-burn method or Swedish cultivation)* where 34% of households are involved in plough cropping only, 19% are involved in *jhum* only, and the rest of them do both plough and *jhum* agriculture [11]. A recent study shows that shifting cultivation is a form of land use among resource-poor communities with a rotation of cultivation and fallow in the same unit of land [12]. *Jhum* farming is changing fast in many regions, owing to population pressures as well as diversification of livelihood patterns to include permanent cash crop production and off-farm labor. However, due to the topographic characteristics, the people of CHT have limited access to livelihood diversification [6]. Recent studies also show that income and non-income poverty is widespread in the southeastern region of Bangladesh, and farmers of this area experience food shortages for two to six months a year [11,13,14].

Nevertheless, modern agricultural practices have significantly changed the crop field by reducing dependency on human labor and saving the farmer's time [15]. The development of high-yielding varieties (HYVs) has flourished in a new era in the agricultural sector [16]. These improved practices are cost-effective as they notably increase the quality and quantity of yield [17]. These practices can help the backward people to move toward a more sustainable living. Acceptance of these technologies will provide farmers with a better subsistence.

Various studies have investigated the impacts of different livelihood activities on rural communities across diverse settings. For instance, Miah et al. [18] underscored the substantial role of shifting cultivation and forest income in the livelihoods of the Chakma tribe in Bangladesh. Tailor et al. [19] pinpointed challenges confronting tribal milk producers in Rajasthan, such as low productivity and limited access to green fodder. Xaba & Masuku [20] analyzed factors affecting profitability in vegetable production, while Patr et al. [21] assessed constraints in traditional pig production systems in Northeast India. Ahmed et al. [22] divided into livelihood diversification patterns in rural Bangladesh, highlighting the significance of rice cultivation and entrepreneurship.

Moreover, Martin & Lorenzen [23] explored the motivations behind livelihood diversification in Laos, linking it to wealth status and asset ownership. Gautam & Andersen [24] investigated how investing in high-return sectors influenced household well-being in Nepal, noting entry barriers leading to inequality. Mariyono [25] examined factors shaping farmers' intentions to commercialize vegetable farming in Indonesia. Punitha et al. [26] identified constraints to livelihood diversification as perceived by jumians in Northeast

India. Jannat et al. [27] scrutinized the impact of agricultural modernization on sustainable livelihoods in Bangladesh, emphasizing the importance of training in new technologies. All of these studies collectively highlight the crucial role of livelihood diversification in enhancing household welfare. While research on income diversification in Bangladesh has been extensive, there is a notable gap in studies focusing on tribal farmers in hilly areas, their agricultural practices, and socioeconomic conditions. Therefore, a study investigating livelihood diversification and its influence on adopting modern agricultural practices in hilly areas is warranted. To shed light on this issue, this study assesses the socioeconomic status of tribal households, evaluates earnings from diverse activities, analyzes production practices, explores the impact of diversification on adopting modern agricultural methods, addresses their challenges, and proposes coping strategies. The overall aim of the study was to find out the profitability of agricultural production practices, the extent of livelihood diversification, and the influence of livelihood diversification on the adoption decisions of modern agricultural practices. The specific objectives were as follows: (a) to depict the socioeconomic status and production practices of tribal farmers; (b) to examine the profitability analysis of crop, livestock production, and homestead gardening; (c) to document the influence of livelihood diversification on the adoption of modern agricultural practices decisions in the hilly areas of Bangladesh; and (d) to identify the problems faced by the respondents and suggest policy options.

## Materials and methods

### Selection of the study area, data sources and sampling technique

The study area was selected to align with the research objectives, which focused on tribal farmers. The CHT region is home to the majority of tribal communities in Bangladesh. Rangamati, one of the districts in CHT, consists of 10 subdistricts, and Baghaichari is one of these upazilas. The foundational maps, showing administrative boundaries and major rivers, were created using datasets from the Bangladesh Country Almanac (BCA) and processed with ArcGIS 10.8.2®. These maps were projected using the WGS1984 coordinate system. The selected subdistrict was Baghaichari is shown in Fig 1.

Distance of Baghaichari is 118 Kilometers from Rangamati Sadar. Primary data were collected directly from farmers, while secondary data sources included authentic and dependable published articles, reports, journals, and documents. To estimate the sample size, the Yemane method was used, yielding a result of 99.97 at a 10% precision level. A total of 115 samples were collected using a simple random sampling technique. Data were collected from the farmers through direct interviews. Appropriate timing of data collection is equally important. The month of February was chosen for data collection when *Boro* rice is grown in the farmer's field.

### Analytical techniques

**Profitability analysis.** The income from farm and non-farm activity was estimated by using profitability equations that determined net return, total variable costs, total fixed cost, benefit-cost ratio (BCR) (Undiscounted), etc. Profit is the difference between the value of goods and services produced by the farm and the costs of resources used in production.

$$NR = GR - (TVC + TFC) \tag{1}$$

where, NR = Total net return (Tk.); GR = Gross return (Tk.); TFC = Total fixed cost (Tk.); and TVC = Total variable cost (Tk.).

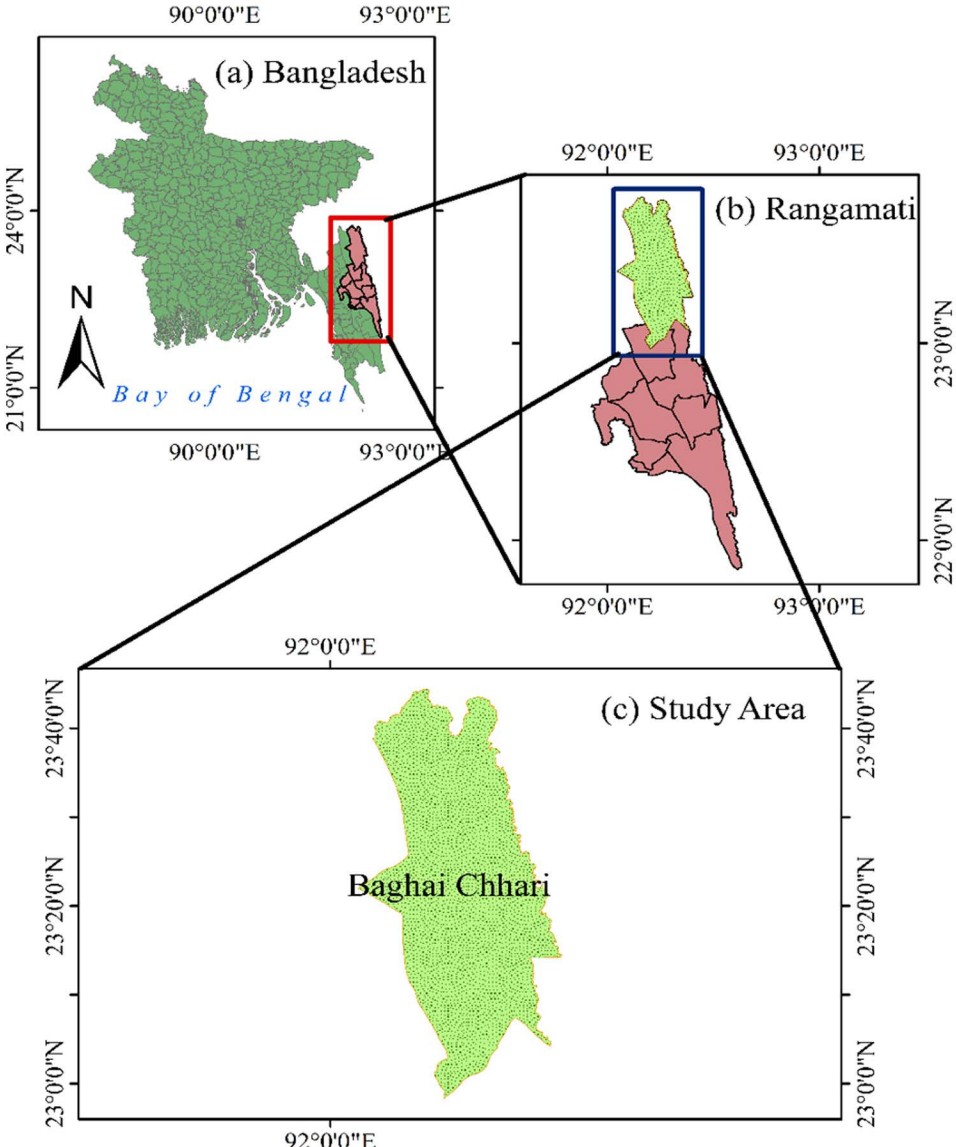

**Fig 1. Maps of the study area.** (a) Map of Bangladesh indicating the location of the study area within the Rangamati district. (b) Map of Rangamati district, highlighting the study region. (c) Close-up of the Baghai Chhari subdistrict, showing the specific study area (in green). Source: Authors' development using ArcGIS 10.8.2® software.

**Simpson index of diversification (SID).** The income diversification sources were analyzed using the Simpson index of diversification ( *SID* ). The formula for the Simpson Index [28] is:

$$SID = 1 - \sum_i P_i^2 n_i \tag{2}$$

where $n$ is the total number of income sources and $p_i$ is the income portion of i-th income source. The value of *SDI* falls between 0 and 1. The index's value is zero if there is just one source of income. As the number of sources increases, the shares ( *Pi* ) decline, as does the sum of the squared shares, so that *SID* approaches 1. Households with the

most diversified income sources have the largest $SID$ value and the least diversified income sources have the smallest $SID$ value. The higher the number of income sources as well as the more evenly distributed the income shares, the higher the value of $SID$. The SID was affected both by the number of income sources as well as by the distribution of income among different sources. Based on the $SDI$ values, the level of livelihood diversification was classified as (a) no diversification ($SDI$ <= 0.01); (b) low level of diversification ($SDI$ = 0.01-0.25); (c) medium level of diversification ($SDI$ = 0.26-0.50); (d) high level of diversification ($SDI$ = 0.51-0.75); and (e) very high level of diversification ($SDI$ =>0.75).

In this study, respondents were asked to indicate the household's major livelihood activities such as crop production, livestock rearing, daily labor, small business, government and non-government jobs, etc. This question helped to understand the livelihood structure at the household level. Following past studies [8], livelihood activities were classified as on-farm livelihood activities comprising crop and livestock production, off-farm activities (which included earning wages for work on other farms), and non-farm activities where income was earned from non-agricultural sources such as job-holding and self-employment.

**Factors influencing the adoption decisions of modern agricultural practices.** Following Wongnaa & Babu [29], the binary logistic regression model was employed in analyzing the influence of livelihood diversification and other socio-economic variables on the adoption of modern agricultural practices decisions of the tribal farmers. Farmers may or may not respond fully or partially to modern agricultural practices or may not respond at all. This led to a binary dependent variable, making the logit model an appropriate approach to be employed in this study.

The adoption model representing adoption behavior employed in this study follows the threshold theory of decision-making, proposed by Hill & Kau [30]. That is individuals with choices to make have reaction thresholds, which are determined by a multiplicity of factors. Choices of this nature are usually modeled as:

$$y_i = \beta_i x_i + u_i \tag{3}$$

where, $x_1$ = Livelihood diversification index (SID); $x_2$ = Age of the respondents in years; $x_3$ = Years of schooling; $x_4$ = Access to extension service; $x_5$ = Access to information; $x_6$ = Access to credit; $x_7$ = Average annual income; $x_8$ = Farming experience; $y_i$ is equal to one if at least one modern agricultural practice is adopted by the farmer and zero otherwise.

$y_i$ = 1, if, $x_i \geq$ x* and $y_i$ = 0, if, xi < x* and x* is a critical value representing the joint effect of independent variables at the threshold level.

The model is binary, which involves estimating and examining the likelihood of adoption of modern agricultural practices, $y$, as a function of independent variables, $x$. For this study $y_i$ therefore takes a value of 1 if a tribal farmer adopts at least one modern agricultural practice, and 0 if he/she does not adopt any of them. The estimation of such a qualitative response model allows us to estimate the conditional probability that $y_i$ assumes one of the specified values. Thus, the probability that a given tribal farmer will adopt at least one adaptation technology, $\beta_i x_i >$ x* ($y_i$ =1) is given as:

$$P_i = Prob(y_i = 1) = F(\beta' x_i) \tag{4}$$

The probability that a tribal farmer will not adopt any modern agricultural practices is, therefore, given as:

$$1 - P_i = Prob\ (y_i = 0) = 1 - F\left(\beta' x_i\right) \tag{5}$$

where, $y_i$ is the observed response of the *ith* observation of the response variable $y$ and $x_i$ is a set of explanatory variables of the *ith* respondent.

**Problems faced by the tribal farmers.** Problems faced by tribal farmers in the study areas were measured using structured questionnaire. Farmers were asked to provide their opinions on selected problems identified during the data collection period. A four-point rating scale was used to compute the constraint score for each respondent, with scores of 3, 2, 1, and 0 assigned to indicate the extent of constraint as high, medium, low, and not at all, respectively. The total constraint scores for each respondent were calculated by summing their scores for all the constraints. The constraint facing index (CFI) was computed using the following formula [31].

$$CFI = \left(C_h \times 3\right) + \left(C_m \times 2\right) + \left(C_l \times 1\right) + \left(C_n \times 0\right) \tag{6}$$

where, $CFI$ = Constraints facing index; $C_h$ = Number of respondents having high constraints; $C_m$ = Number of respondents having medium constraints; $C_l$ = Number of respondents having low constraints; and $C_n$ = Number of respondents having no constraints.

## Ethical approval

Ethical approval was sought and received from the Sylhet Agricultural University Research System (SAURES) in accordance with the rules and regulations of the Sylhet Agricultural University, Sylhet, Bangladesh. The type of approval was exempt because participation in the study did not expose the respondents to danger.

## Results and discussion

### Socioeconomic characteristics of the tribal farmers

Table 1 shows that most of the farmers (64.35%) belonged to the 15-49 age group, whereas 41 respondents (35.65%) belonged to the above 49 age category, and none was below 15. The findings demonstrate that most of the respondents were male (70.43%). The average family size is 4.91 in the study area, which is slightly higher than the country's average rural area household size of 4.11 [32]. It was found that most of the families have 1-5 members (67.83%), and the rest, 32.17%, belong to the 6-10 family members category. In the study area, most people took credit from the Bangladesh government social welfare program, '*Ektee Bari Ektee Khamar*', and a few took out loans from banks.

These loans were primarily used for agricultural activities (Purchasing inputs such as seeds, fertilizers, and pesticides, renting machinery) and non-agricultural income generating activities (establishing small businesses such as shops or handicrafts, expanding non-farm enterprises like homestead gardening or livestock rearing). Data shows that only 34.72% of tribal farmers can get extension services from the extension personnel. In the study area, 84.35% of respondents got access to the right information. Table 1 also shows that most respondents are marginal farmers (51.30%).

In this study, all the respondents were crop cultivators. Besides that, their occupational pattern is significantly diversified. Table 1 demonstrates that 34.78% of farmers were engaged in Crop cultivation + Poultry and Livestock rearing. The number of sources of income is an important indicator of the livelihood diversification of a particular household. Previous researchers have also used this technique [33]. Households having more

**Table 1. Basic socioeconomic characteristics of the tribal farmers.**

| Characteristics | No. of respondents | Percentage of respondents (%) |
|---|---|---|
| **Age categories** | | |
| <15 | 0 | 0.0 |
| 15-49 | 74 | 64.35 |
| >50 | 41 | 35.65 |
| **Sex** | | |
| Male | 81 | 70.43 |
| Female | 34 | 29.56 |
| **Family size** | | |
| 1-5 | 78 | 67.83 |
| 6-10 | 37 | 32.17 |
| >10 | 0 | 0 |
| **Level of education** | | |
| Illiterate | 46 | 40 |
| Primary | 35 | 30.43 |
| Secondary | 20 | 17.4 |
| Above secondary | 14 | 12.17 |
| **Access to credit** | | |
| No credit access | 68 | 59.13 |
| Credit access | 47 | 40.87 |
| **Access to extension service** | | |
| Had access | 40 | 34.78 |
| No access | 75 | 65.21 |
| **Access to information** | | |
| Had access | 97 | 84.35 |
| No access | 18 | 15.65 |
| **Farm size** | | |
| Landless (<.2ha) | 18 | 15.65 |
| Marginal (.2ha-.6ha) | 59 | 51.30 |
| Small (.6ha-1.008ha) | 34 | 29.56 |
| Medium (1.008ha-3.04ha) | 4 | 3.48 |
| Large (>3.04ha) | 0 | 0 |
| **Occupational patterns** | | |
| Crop cultivation | 11 | 9.57 |
| Crop cultivation + Livestock rearing | 11 | 9.57 |
| Crop cultivation + non-farm | 10 | 8.69 |
| Crop cultivation + Poultry and livestock rearing | 40 | 34.78 |
| Crop cultivation + Poultry rearing + non-farm | 8 | 6.95 |
| Crop cultivation + Poultry and Livestock rearing + non-farm | 19 | 16.52 |
| Crop cultivation + Livestock rearing + non-farm | 16 | 13.91 |
| **Sources of income diversification** | | |
| <0.01 (no diversification) | 0 | 0 |
| 0.01-0.25 (low level) | 17 | 14.78 |
| 0.26-0.50 (medium level) | 28 | 24.35 |
| 0.51-0.75 (high level) | 61 | 53.04 |
| >0.75 (very high level) | 9 | 7.82 |

Source: Author's estimation based on field survey, 2021.

sources of income are more diversified. In our study, we defined livelihood diversification as the extent to which households engage in multiple IGAs beyond their primary occupation. Households with more sources of income, such as combining crop farming with livestock rearing, non-farm businesses, or wage labor, are inherently more diversified due to their engagement in varied economic activities. This approach aligns with the established definitions of diversification in the livelihood literature, where diversification is measured by the variety and number of income streams [34,35]. Diversification is measured using the SID, which is categorized into five groups; up to 0.01 is considered no diversification. Between 0.01-0.25 is low, 0.26-0.50 is medium, 0.51-0.75 is high, and more than 0.75 is a very high level of diversification [22]. The average SID was 0.588. Table 1 also shows that most respondents (53.04%) were in a high-level diversified group. There was no respondent with zero SID value. About 28 respondents (24.35%) had a medium level of diversification. Of the sampled households, 14.78% had a low, and 7.82% had a very high level of livelihood diversification.

## Average annual income and expenditure of the tribal farmers

Income is the most important indicator of people's socioeconomic status. The annual income of a family was estimated based on all the year-round, and this study categorized the income of the household by the classification of Islam et al. [36]. The average total family income was calculated by summing up the farm and non-farm sources of income during the study period. Fig 2 shows that 4.34% of respondents were classified as low-income people, 10.43% were medium-income, and 85.22% were high-income.

Average annual expenditure is another important factor to indicate the socioeconomic status of the respondents. The average annual expenditure of the respondents was classified into three groups, i.e., low, medium, and high expenditure. Fig 3 shows that around 6.08% of the respondents were in the low-expenditure category, 8.69% were in the medium-expenditure category, and 85.21% were in the high-expenditure category.

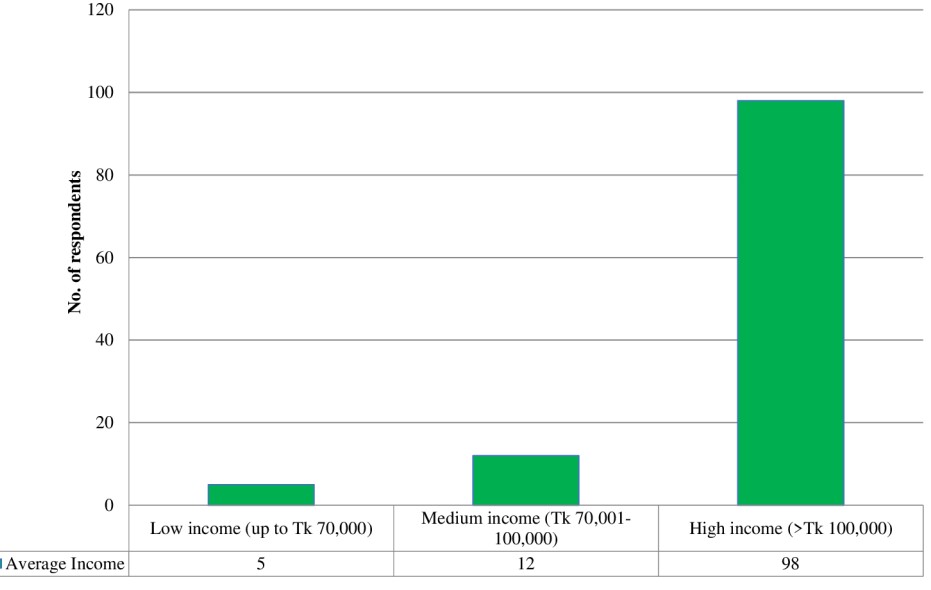

**Fig 2. Distribution of average annual income of the respondents.**

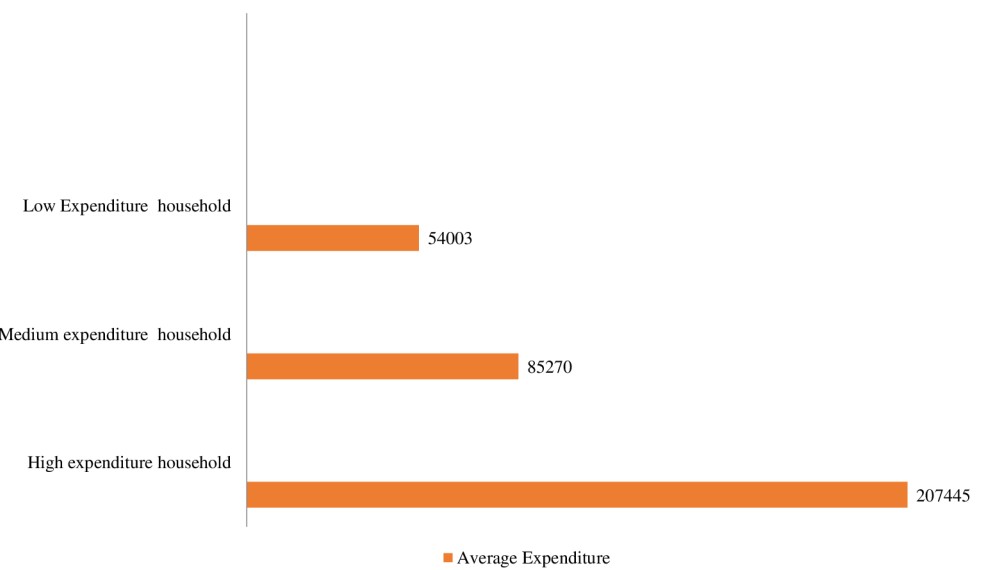

**Fig 3. The average annual expenditure of the respondents.**

The majority of the respondents fell into the high-expenditure category. Food items were the primary expenditure items of the medium and high categories of expenditure groups. The total expenditures for low, medium, and higher expenditure respondents were Tk. 54003 (USD 646.74), Tk. 85270 (USD 1021.19), and Tk. 207445 (USD 2484.37), respectively.

## Adoption status of modern agricultural practices

Table 2 shows that tribal farmers have also adopted modern technologies, which have increased their production, reduced costs, and made crop production more profitable. HYVs are high-yielding, which attracted the farmers most (93.04%). Nearly 94.78% of farmers applied pesticides, and 89.57% used fertilizers. However, a small percentage of farmers (5.22%) employed an environment-friendly pest control method called 'perching.' which incurs minimal to no cost.

**Table 2. Adoption status of modern agricultural practices.**

| Items | Adopted | Not adopted |
| --- | --- | --- |
| HYVs | 107 (93.04) | 8 (6.96) |
| Use of pesticides | 109 (94.78) | 6 (5.22) |
| Perching method | 6 (5.22) | 109 (94.78) |
| Use of chemical fertilizers | 103 (89.57) | 12 (10.43) |
| Artificial irrigation | 86 (74.78) | 29 (25.22) |
| Artificial dams | 57 (49.57) | 58 (50.43) |
| Use of tractors | 103 (89.57) | 12 (10.43) |
| Use of reaper | 85 (73.91) | 30 (26.09) |
| Seed fertilizer drills | 0 (0.00) | 115 (100.00) |
| Internet and apps | 6 (5.22) | 109 (94.78) |

Source: Author's estimation based on field survey, 2021.

Note: The figure in the parenthesis indicates the percentage.

Chemical pesticides are comparatively expensive. Water pumps play a crucial role in irrigation in the study area. About 73.91% of farmers also use mechanical harvest through reapers, which reduces their harvesting cost and risk of crop damage and boosts farmers' efficiency. No farmers were using seed fertilizer drills. Fruit orchard owners relied on the internet and used specialized apps to manage plant diseases. Commercial producers, in particular, adopted advanced fruit varieties.

## Profitability analysis

**Cost and return analysis of different agricultural components.** Table 3 depicts the total cost of rice production per hectare is Tk. 76220.94 (USD 912.82), with the total variable cost at Tk. 62813.56 (USD 752.25), and the total fixed cost at Tk. 13407.38 (USD 160.56). The Gross return from one hectare of land is Tk. 111248.5 (USD 1332.31). The gross margin per hectare of the rice production is Tk. 48434 (USD 580.05), and the net margin is Tk. 35027.56 (USD 419.49). Finally, the BCR (undiscounted) for rice production per hectare is 1.46, implying that 1.46 would be earned by every Tk. 1 investing in rice production.

In livestock rearing, variable costs include feed costs, medicine facilities costs, artificial insemination costs, casual labor costs, etc. The total cost of rearing livestock per annum in the study area is Tk. 41614.16 (USD 498.37), where the total fixed cost is Tk. 6715.96 (USD 80.43), and the total variable cost is Tk. 34898.2 (USD 417.94). It can be seen from Table 3 that the annual gross return of livestock rearing is Tk. 58740.27 (USD 703.47). The analysis shows that the gross margin of rearing livestock per annum is Tk. 23842.07 (USD 285.53). The annual net return of the sampled farmers for livestock rearing is Tk. 17126.11 (USD 205.10). Finally, the BCR for rearing livestock in the study area is 1.41.

*Jhum* cultivation is an ancient method; however, it's destructive to the environment. Several research has shown that agroforestry can be a better alternative to *jhum* cultivation. Table 3 represents the profitability of homestead enterprise which includes both vegetable farming and fruit orchards. The total variable cost for homestead gardening is Tk. 42,308.09 (USD 506.68) per year, which covers expenses such as seeds, fertilizers, labor, and other inputs that fluctuate with the scale of production. The total fixed cost for homestead gardening is Tk.

**Table 3. Profitability analysis of different agricultural components.**

| Cost and return items | Rice production (Tk./Ha) | Livestock rearing (Tk./Year) | Homestead gardening (Tk./Year) |
|---|---|---|---|
| Total variable cost (A) | 62813.56 (752.25) | 34898.2 (417.94) | 42308.09 (506.68) |
| Total fixed cost (B) | 13407.38 (160.56) | 6715.96 (80.43) | 14476.34 (173.36) |
| Total cost (C) | 76220.94 (912.82) | 41614.16 (498.37) | 56784.43 (680.05) |
| Gross return (D) | 111248.5 (1332.31) | 58740.27 (703.47) | 61933.45 (741.71) |
| Gross margin (D-A) | 48434.94 (580.05) | 23842.07 (285.53) | 19625.36 (235.03) |
| Net return (E-B) | 35027.56 (419.49) | 17126.11 (205.10) | 5149.02 (61.66) |
| BCR (Undiscounted) | 1.46 | 1.41 | 1.09 |

Source: Author's estimation based on field survey, 2021.

Note: The figures in parentheses indicate the value in USD.

Note: 1 USD = Tk. 83.5 (conversion rate based on the average of September 2021).

14,476.34 (USD 173.36), accounting for expenses such as land preparation and equipment depreciation, which do not vary with production levels. The gross return from homestead gardening in the study area is Tk. 61,933.45 (USD 741.71) per year, reflecting the revenue generated from the sale of vegetables and fruits. This yields a gross margin of Tk. 19,625.36 (USD 235.03), indicating the financial benefit after covering the variable costs associated with the activity. The net return from homestead gardening is Tk. 5,149.02 (USD 61.66) annually, which represents the profit remaining after both variable and fixed costs are deducted.

The BCR for homestead gardening is 1.09, meaning that for every Tk. 1 invested in homestead gardening, the farmer earns Tk. 1.09. While the profitability of homestead gardening is lower than that of rice production and livestock rearing, it still provides a positive return, demonstrating its value as an income-generating activity.

**Factors influencing the adoption decision of modern agricultural practices.** Table 4 presents the results of a logistic regression analysis investigating the factors influencing tribal farmers' adoption decision of modern agricultural practices. The table reports the coefficients, standard errors (S.E.), Wald statistics, p-values, and marginal effects (ME) for each of the explanatory variables considered in the model. Livelihood diversification ($p < 0.05$) is significantly associated with adopting modern practices. The marginal effect shows that a one-unit increase in diversification value (SID) increases the probability of adopting modern agricultural practices by 1.14 unit. As diversification helps increase the probability of livelihood sustainability, adopting modern practices becomes easy for the farmers.

The age of the farmer had a significant correlation with the adoption decision ($p < 0.05$). The marginal effect of age indicates that a one unit increase in the farmer's age decreases the probability of adopting modern agricultural practices by 0.88 unit. This could be because young people are more concerned about adopting new technologies. As a result, a one-year increase in age decreases the probability of adoption in this context. Our findings are in line with previous studies, and it is evidenced that the age of the farmers significantly influences the adoption of agricultural technologies [37–41]. Access to information ($p < 0.05$) was found to have a positive and significant correlation with modern practices adoption in agriculture. The marginal effect shows that access to information increases the probability of adopting

Table 4. Factors influencing the adoption decisions of modern agricultural practices.

| Variables | Coefficient | S.E. | Wald | p-value | ME |
|---|---|---|---|---|---|
| Constant | -2.634 | 0.749 | 11.839 | 0.001 | 0.076 |
| Livelihood diversification (SID) | 0.205** | 0.037 | 6.903 | 0.034 | 1.14 |
| Age of the respondent | -0.031** | 0.010 | 4.244 | 0.039 | 0.879 |
| Years of schooling | 0.028 | 0.048 | 0.635 | 0.425 | 1.029 |
| Extension service (yes = 1) | 0.041 | 0.423 | 0.040 | 0.675 | 0.945 |
| Access to information (yes = 1) | 0.777** | 0.256 | 4.213 | 0.030 | 2.134 |
| Access to credit (yes = 1) | -0.067 | 0.345 | 0.056 | 0.767 | 0.734 |
| Average annual income | 0.452** | 0.112 | 14.600 | 0.024 | 1.932 |
| Farming experience | 0.067*** | 0.020 | 14.344 | 0.000 | 1.343 |
| Hosmer and Lemeshow Test | Chi-square = 4.260; p-value: 0.833 | | | | |

Note: ME = Marginal Effect,

*Significance level of 10%,

**Significance level of 5%,

***Significance level of 1%

Source: Author's estimation based on field survey, 2021.

modern technology 2.13 times. The reason could be access to information highly influences farmers to switch to modern technologies. This is in line with the findings of Sarker et al. [40], who found that access to information through extension personnel and farmers' field schools (FFS) would positively influence the adoption of new agricultural technologies. Along with this, Wu [42] found that accessing information online has a significant positive impact on family farms' adoption of new technologies.

The household income of a particular household is significantly associated with adoption decisions at a 5% significant level. The marginal effect of income indicates that a unit increase in income increases the probability of adopting modern agricultural practices by 1.93 unit. Increased household income increases the affordability of adopting modern agricultural practices. This study reveals that the average annual income of farmers is significantly associated with adoption decisions. Higher-income levels generally provide farmers with the financial resources needed to invest in new technologies, purchase inputs, and manage risks associated with adopting innovative practices. Our findings are in line with previous studies, and it is evidenced that the farmers' average annual income inspires them to adopt new farm technologies [42]. Farming experience ($p < 0.01$) is significantly associated with the adoption of modern agricultural practices. The marginal effect indicates that a one unit increase in farming experience increases the probability of adopting modern practices by 1.34 unit. There may be a reason behind this: with the increase in farming experience, farmers get a better understanding of the efficiency of different practices. The value of the chi-square test is 4.26, and the estimated p-value is 0.83, which is insignificant. It means the model is well-fitted. This finding is in line with a previous study, and it is evidenced that the farming experience of the farmers positively affects modern technology adoption [40,41]. However, Ghosh et al. [43] found no significant relationship between farming experience and the adoption of modern agricultural practices.

The results suggest that livelihood diversification (SID), age of the respondent, access to information, average annual income, and farming experience are key factors influencing the adoption decision of modern agricultural practices among tribal farmers in Bangladesh. Livelihood diversification and farming experience are positively correlated with the adoption of modern farming techniques, while older farmers are less likely to adopt these practices. Access to information and income also play significant roles in adoption decisions. These findings highlight the importance of improving information dissemination, supporting livelihood diversification, and addressing income disparities to promote the adoption of modern agricultural practices in rural areas.

**Problems faced by the tribal farmers.** In the study area, the computed value of CFI for the major problem was 238 out of a possible 345, with 23 respondents facing it to a high extent, 76 to a medium extent, and 16 to a low extent. The second-highest constraint was the product's price during harvesting season, affected by an excessive supply that reduced prices to Tk. 19-20 per kg of paddy. High-interest rates were also significant, with a CFI of 228, ranked fourth, and affecting 21 farmers to a high extent, 73 to a medium extent, and 19 to a low extent. Small and fragmented land holdings, which discourage the adoption of new practices, were another issue with a CFI of 215.

Table 5 highlights that the most significant constraints for tribal farmers are related to financial access (e.g., credit supply, interest rates) and market conditions (e.g., low product prices, and lack of marketing facilities). Additionally, limited access to technology and environmental challenges like natural calamities and disease outbreaks further compound the difficulties faced by farmers. These findings suggest that addressing these key constraints through improved financial services, better market access, and increased support for technology adoption could significantly enhance the livelihoods of tribal farmers and encourage the adoption of modern agricultural practices.

**Table 5. Constraints faced by the tribal farmers.**

| Constraints | CFI | Rank |
|---|---|---|
| Lack of sufficient credit supply | 232 | 3 |
| High-interest rate | 228 | 4 |
| Lack of knowledge about modern agricultural practices | 97 | 10 |
| Low price of product during harvesting | 237 | 2 |
| Fragmented land | 215 | 5 |
| Shortage of human labor during peak period | 0 | 13 |
| Lack of suitable marketing facilities | 214 | 6 |
| Poor access to new technologies | 238 | 1 |
| Dominance of intermediaries | 167 | 8 |
| Disease attacks | 135 | 9 |
| Loss of product due to theft | 24 | 12 |
| The burden of old debt | 71 | 11 |
| Natural calamities | 175 | 7 |

Source: Author's estimation based on field survey, 2021.

## Conclusion and policy implications

The study explored the role of livelihood diversification in shaping tribal farmers' decisions to adopt modern agricultural practices while also assessing the profitability of various agricultural practices and identifying key constraints and policy recommendations to enhance the sustainability of tribal farming livelihoods in Bangladesh. The BCR for rice production, livestock rearing, and homestead gardening was 1.46, 1.41, and 1.09, respectively, with rice production showing the highest profitability. The regression results indicated that the decision to adopt modern agricultural practices was significantly correlated with livelihood diversification, the age of the respondents, access to information, household income, and farming experience. Major constraints faced by tribal farmers included poor access to modern practices, low harvest prices, and insufficient credit supply. Livelihood diversification emerged as crucial for sustainable livelihoods, improving household income and resilience. Modern agricultural practices have boosted crop productivity, supporting food security. Thus, promoting livelihood diversification among tribal farmers is essential for improving their resilience and income sustainability. This can be achieved by offering training, market access, and financial support for activities like poultry and livestock rearing, as well as non-farm enterprises such as homestead gardening. Policy recommendations include encouraging rice production due to its profitability, promoting off-farm income activities, ensuring access to quality inputs at reasonable prices, and facilitating access to credit with lower interest rates. Better communication infrastructure in geographically diverse areas like the CHT is essential for marketing facilities. Measures to reduce cost of production and enhance market information dissemination are also proposed. Extension services should assist farmers in adopting modern agricultural technologies, while encouragement of off-farm activities can provide a more robust livelihood structure.

## Supporting information

**S1 File. All the variables data of the survey sample.**
(XLSX)

## Author contributions

**Conceptualization:** Anika Tasnim, Jasim Uddin Ahmed, Md. Jahangir Alam Khan, Md. Abdullah Al-Noman, Arifa Jannat, Md. Monirul Islam.

**Data curation:** Anika Tasnim, Jasim Uddin Ahmed, Md. Jahangir Alam Khan.

**Formal analysis:** Anika Tasnim, Arifa Jannat, Md. Monirul Islam.

**Investigation:** Jasim Uddin Ahmed, Md. Jahangir Alam Khan, Md. Monirul Islam.

**Methodology:** Anika Tasnim, Md. Abdullah Al-Noman, Arifa Jannat, Md. Monirul Islam.

**Project administration:** Jasim Uddin Ahmed, Md. Jahangir Alam Khan.

**Resources:** Jasim Uddin Ahmed, Md. Jahangir Alam Khan, Md. Abdullah Al-Noman, Arifa Jannat, Md. Monirul Islam.

**Software:** Arifa Jannat, Md. Monirul Islam.

**Supervision:** Jasim Uddin Ahmed, Md. Monirul Islam.

**Validation:** Md. Abdullah Al-Noman, Arifa Jannat, Md. Monirul Islam.

**Visualization:** Md. Abdullah Al-Noman, Arifa Jannat, Md. Monirul Islam.

**Writing – original draft:** Anika Tasnim, Jasim Uddin Ahmed, Md. Jahangir Alam Khan, Md. Abdullah Al-Noman, Arifa Jannat, Md. Monirul Islam.

**Writing – review & editing:** Anika Tasnim, Jasim Uddin Ahmed, Md. Jahangir Alam Khan, Md. Abdullah Al-Noman, Arifa Jannat, Md. Monirul Islam.

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
