## [Decision Letter · Decision Letter 0]

20 Nov 2024

PONE-D-24-36026Influence of livelihood diversification on the adoption of modern agricultural practices by tribal farmers in hilly areas of BangladeshPLOS ONE

Dear Dr. Islam,

Thank you for submitting your manuscript to PLOS ONE. After careful consideration, we feel that it has merit but does not fully meet PLOS ONE’s publication criteria as it currently stands. Therefore, we invite you to submit a revised version of the manuscript that addresses the points raised during the review process.

We look forward to receiving your revised manuscript.

Kind regards,

Meraj Alam Ansari

Academic Editor

PLOS ONE

Journal Requirements:

2. We note that your Data Availability Statement is currently as follows: [All relevant data are within the manuscript and its Supporting Information files.] Please confirm at this time whether or not your submission contains all raw data required to replicate the results of your study. Authors must share the “minimal data set” for their submission. PLOS defines the minimal data set to consist of the data required to replicate all study findings reported in the article, as well as related metadata and methods (https://journals.plos.org/plosone/s/data-availability#loc-minimal-data-set-definition). For example, authors should submit the following data: - The values behind the means, standard deviations and other measures reported; - The values used to build graphs; - The points extracted from images for analysis. Authors do not need to submit their entire data set if only a portion of the data was used in the reported study. If your submission does not contain these data, please either upload them as Supporting Information files or deposit them to a stable, public repository and provide us with the relevant URLs, DOIs, or accession numbers. For a list of recommended repositories, please see https://journals.plos.org/plosone/s/recommended-repositories. If there are ethical or legal restrictions on sharing a de-identified data set, please explain them in detail (e.g., data contain potentially sensitive information, data are owned by a third-party organization, etc.) and who has imposed them (e.g., an ethics committee). Please also provide contact information for a data access committee, ethics committee, or other institutional body to which data requests may be sent. If data are owned by a third party, please indicate how others may request data access.

Additional Editor Comments:

Thank you for submission of your manuscript in the PloS One journal. Based on our review as well as reviewer’s comments and suggestion, your manuscript is required major revision. The comments and suggestions are given in the reviewer’s comment.

Thank you

Reviewers' comments:

Reviewer's Responses to Questions

**Comments to the Author**

1. Is the manuscript technically sound, and do the data support the conclusions?

Reviewer #1: Yes

Reviewer #2: Yes

2. Has the statistical analysis been performed appropriately and rigorously? 

Reviewer #1: Yes

Reviewer #2: Yes

3. Have the authors made all data underlying the findings in their manuscript fully available?

Reviewer #1: Yes

Reviewer #2: No

4. Is the manuscript presented in an intelligible fashion and written in standard English?

Reviewer #1: Yes

Reviewer #2: Yes

5. Review Comments to the Author

Reviewer #1: The study explored livelihood diversification and its impact on the adoption of modern agricultural techniques. The results indicated that the decision to adopt modern agricultural practices was significantly correlated with livelihood diversification, the age of the farmers, access to information, household income, and farming experience. Major constraints faced by tribal farmers included poor access to modern practices, low harvest prices, and insufficient credit supply. Livelihood diversification emerged as crucial for sustainable livelihoods, improving household income and resilience. Modern agricultural practices have boosted crop productivity, supporting food security. Thus, promoting livelihood diversification among tribal farmers is essential for improving their resilience and income sustainability. This can be achieved by offering training, market access, and financial support for activities like poultry and livestock rearing, as well as non-farm enterprises such as homestead gardening. Policy recommendations include encouraging rice production due to its profitability, promoting off-farm economic activities, ensuring access to quality inputs at reasonable prices, and facilitating access to credit with lower interest rates.

Overall article is written well, however, english language need to be improved.

Reviewer #2: The paper can be accepted after the revision. The detailed comments are provided inside the documents.

The authors are suggested to give line number while revising the paper. It is difficult to provide the comments .

The author needs to include the data availability consent for crosschecking the data used in the manuscript.

The research content of the manuscript seems to deviate from the goal in few places especially in the discussion part. Please revised in accordingly. .

6. PLOS authors have the option to publish the peer review history of their article (what does this mean? ). If published, this will include your full peer review and any attached files.

**Do you want your identity to be public for this peer review?** For information about this choice, including consent withdrawal, please see our Privacy Policy .

Reviewer #1: No

Reviewer #2: No

---

## [Author Response · Author response to Decision Letter 0]

28 Nov 2024

The Role of Livelihood Diversification in Agricultural Modernization Among Tribal Farmers in Bangladesh: A Binary Logistic Model Approach

Manuscript ID: PONE-D-24-36026

Responses to Reviewer’s Comments

Thank you so much for considering the paper based on the proposed revisions. The authors gave full consideration of making the corrections advised after thoroughly reading the comments. The point-to-point questions and answers are given below. Besides, please note that the changed areas are based on the reviewer's comments.

Reviewer-1: Comments

The study explored livelihood diversification and its impact on the adoption of modern agricultural techniques. The results indicated that the decision to adopt modern agricultural practices was significantly correlated with livelihood diversification, the age of the farmers, access to information, household income, and farming experience. Major constraints faced by tribal farmers included poor access to modern practices, low harvest prices, and insufficient credit supply. Livelihood diversification emerged as crucial for sustainable livelihoods, improving household income and resilience. Modern agricultural practices have boosted crop productivity, supporting food security. Thus, promoting livelihood diversification among tribal farmers is essential for improving their resilience and income sustainability. This can be achieved by offering training, market access, and financial support for activities like poultry and livestock rearing, as well as non-farm enterprises such as homestead gardening. Policy recommendations include encouraging rice production due to its profitability, promoting off-farm economic activities, ensuring access to quality inputs at reasonable prices, and facilitating access to credit with lower interest rates.

Overall article is written well, however, the English language needs to be improved.

Response: Thank you for your appreciation of our manuscript. We acknowledge your comment regarding the need to improve the English language. Accordingly, we have thoroughly reviewed the manuscript to enhance the grammar, sentence structure, and overall readability.

Reviewer-2: Comments

The paper can be accepted after the revision. The detailed comments are provided inside the documents.

The authors are suggested to give line numbers while revising the paper. It is difficult to provide the comments.

Response: Thank you for your valuable feedback on our manuscript. We greatly appreciate the time and effort you have invested in reviewing our work. Regarding your suggestion to include line numbers in the revised version of the manuscript, we have ensured that line numbers are now added throughout the document to facilitate easier reference and comment. We apologize for any inconvenience caused by their absence in the original submission.

The author needs to include the data availability consent for crosschecking the data used in the manuscript.

Response: In response to your suggestion regarding data availability consent, we have now included a dedicated Data Availability Statement section in the manuscript. This section provides details about how and where the data used in our study can be accessed for verification and further research.

The research content of the manuscript seems to deviate from the goal in a few places, especially in the discussion part. Please revise accordingly.

Response: Thank you for your constructive feedback on our manuscript, particularly regarding the alignment of the research content with the study’s goals. We appreciate your valuable observation and have carefully revised the manuscript to address this concern. In the revised manuscript, we have thoroughly reviewed and refined the discussion section to ensure it remains focused on the study’s objectives. We have streamlined the arguments to directly address the core research questions, emphasizing the relationship between livelihood diversification, modern agricultural practices, and their impact on sustainability and resilience. We revised the discussion to clearly connect key findings, such as the role of livelihood diversification in sustainable agriculture, with actionable implications and policy recommendations.

Additional Comments from the Manuscript:

Page 13: Loan for what?

Response: Thank you for your insightful observation regarding the source and purpose of loans mentioned in the manuscript.

In response to your query about the specific purposes for which loans were taken, we have clarified this in the revised manuscript. In the study area, most people utilized credit from the Bangladesh government’s social welfare program, Ektee Bari Ektee Khamar (One House One Farm), and a smaller proportion took out loans from banks. These loans were primarily used for agricultural activities (Purchasing inputs such as seeds, fertilizers, and pesticides, renting machinery) and non-Non-Agricultural Income-Generating Activities (establishing small businesses such as shops or handicrafts, expanding non-farm enterprises like homestead gardening or livestock rearing).

Page 14: may be farm family with more diversified option earn more income

Response: Thank you for your valuable observation regarding the relationship between income and livelihood diversification.

In our study, we defined livelihood diversification as the extent to which households engage in multiple income-generating activities beyond their primary occupation. Households with more sources of income, such as combining crop farming with livestock rearing, non-farm businesses, or wage labor, are inherently more diversified due to their engagement in varied economic activities. This approach aligns with the established definitions of diversification in the livelihood literature, where diversification is measured by the variety and number of income streams. We have revised the manuscript to clarify this reasoning and have also elaborated on the potential causal pathways connecting diversification, income stability, and overall household earnings.

Page 15: Can the authors convers TK to USD as this is an International Journal or somewhere mention 1Tk is how much USD

Response: Thank you for your suggestion regarding the conversion of Bangladeshi Taka (Tk) to USD. To improve the manuscript’s clarity for an international audience, we have taken the following steps:

Wherever monetary values are mentioned, we have provided their equivalent in USD using the prevailing exchange rate during the study period. For instance, the conversion rate (e.g., "1 USD = BDT 83.5") has been included in the revised version of the manuscript to serve as a reference.

Page 16: Rephrase this sentence "The perching method hardly costs anything"

Response: Thank you for your feedback. In response to your comment, we have revised the sentence to better convey the intended meaning. The revised sentence now reads: "The perching method incurs minimal to no cost." This revision clarifies that the method is virtually cost-free or requires negligible expenditure.

Page 17: Please consider revising the unit to USD

Response: Thank you for your suggestion regarding the conversion of Bangladeshi Taka (Tk) to USD. To improve the manuscript’s clarity for an international audience, we have taken the following steps:

Wherever monetary values are mentioned, we have provided their equivalent in USD using the prevailing exchange rate during the study period. For instance, the conversion rate (e.g., "1 USD = BDT 83.5") has been included in the revised version of the manuscript to serve as a reference.

---

## [Decision Letter · Decision Letter 1]

16 Jan 2025

The Role of Livelihood Diversification in Agricultural Modernization Among Tribal Farmers in Bangladesh: A Binary Logistic Model Approach

PONE-D-24-36026R1

Dear Dr. Islam,

We’re pleased to inform you that your manuscript has been judged scientifically suitable for publication and will be formally accepted for publication once it meets all outstanding technical requirements.

Kind regards,

Meraj Alam Ansari

Academic Editor

PLOS ONE

Additional Editor Comments (optional):

Reviewers' comments:

Reviewer's Responses to Questions

**Comments to the Author**

1. If the authors have adequately addressed your comments raised in a previous round of review and you feel that this manuscript is now acceptable for publication, you may indicate that here to bypass the “Comments to the Author” section, enter your conflict of interest statement in the “Confidential to Editor” section, and submit your "Accept" recommendation.

Reviewer #3: All comments have been addressed

Reviewer #4: (No Response)

2. Is the manuscript technically sound, and do the data support the conclusions?

Reviewer #3: Yes

Reviewer #4: Yes

3. Has the statistical analysis been performed appropriately and rigorously? 

Reviewer #3: Yes

Reviewer #4: Yes

4. Have the authors made all data underlying the findings in their manuscript fully available?

Reviewer #3: Yes

Reviewer #4: No

5. Is the manuscript presented in an intelligible fashion and written in standard English?

Reviewer #3: Yes

Reviewer #4: Yes

6. Review Comments to the Author

Reviewer #3: The authors have addressed by the comments raised by me. The paper now looks fine for the publication provided it meets all the policy of the journal.

Reviewer #4: Overall comments

1. The article is well-structured and informative; however, there is scope for improvement in the use of English language for better clarity and readability.

2. The discussion part requires more focus; consider adding additional reviews to enhance clarity and depth.

3. Convert the Tk unit to USD for consistency.

4. Overall, the article is good and can be accepted after suggested revisions.

7. PLOS authors have the option to publish the peer review history of their article (what does this mean? ). If published, this will include your full peer review and any attached files.

**Do you want your identity to be public for this peer review?** For information about this choice, including consent withdrawal, please see our Privacy Policy .

Reviewer #3: No

Reviewer #4: **Yes: ** Mohd. Arif

---

## [Editor Report · Acceptance letter]

PONE-D-24-36026R1

PLOS ONE

Dear Dr. Islam,

I'm pleased to inform you that your manuscript has been deemed suitable for publication in PLOS ONE. Congratulations! Your manuscript is now being handed over to our production team.

Kind regards,

on behalf of

Dr. Meraj Alam Ansari

Academic Editor

PLOS ONE